# Risk Factors Affecting Traffic Accidents at Urban Weaving Sections: Evidence from China

**DOI:** 10.3390/ijerph16091542

**Published:** 2019-05-01

**Authors:** Xinhua Mao, Changwei Yuan, Jiahua Gan, Shiqing Zhang

**Affiliations:** 1School of Economics and Management, Chang’an University, Xi’an 710064, China; changwei@chd.edu.cn; 2Department of Civil and Environmental Engineering, University of Waterloo, Waterloo, ON N2L 3G1, Canada; 3Transport Planning and Research Institute, Ministry of Transport, Beijing 100028, China; ganjh@tpri.org.cn; 4School of Management Engineering, Zhengzhou University of Aeronautics, Zhengzhou 450046, China; zshiqing_chd@163.com

**Keywords:** traffic accidents, risk factors, weaving section, multinomial logistic regression

## Abstract

As a critical configuration of interchanges, the weaving section is inclined to be involved in more traffic accidents, which may bring about severe casualties. To identify the factors associated with traffic accidents at the weaving section, we employed the multinomial logistic regression approach to identify the correlation between six categories of risk factors (drivers’ attributes, weather conditions, traffic characteristics, driving behavior, vehicle types and temporal-spatial distribution) and four types of traffic accidents (rear-end, side wipe, collision with fixtures and rollover) based on 768 accident samples of an observed weaving section from 2016 to 2018. The modeling results show that drivers’ gender and age, weather condition, traffic density, weaving ratio, vehicle speed, lane change behavior, private cars, season, time period, day of week and accident location are important factors affecting traffic accidents at the weaving section, but they have different contributions to the four traffic accident types. The results also show that traffic density of ≥31 vehicle/100 m has the highest risk of causing rear-end accidents, weaving ration of ≥41% has the highest possibility to bring about a side wipe incident, collision with fixtures is the most likely to happen in snowy weather, and rollover is the most likely incident to occur in rainy weather.

## 1. Introduction

The weaving section is a common type of road configuration, which widely exists in freeway interchanges [1]. It is formed when a merging area is closely followed by a diverging area, typically within less than 0.76 km [2]. Based on the minimum number of lane changes required for completing the weaving behavior, weaving areas can be grouped into three major types, i.e., Type A, Type B, and Type C, shown in Figure 1.

The three types of weaving sections are defined in detail as follows [2].
Type A weaving section: Every weaving vehicle has to make at least one lane change in the weaving area.Type B weaving section: One weaving movement can be made without making any lane change, while the other weaving movement requires at most one lane change.Type C weaving section: One weaving movement can be made without making any lane change, while the other weaving movement requires at least two lane changes.

In reality, it is also possible that two of these weaving section types can overlap [3]. Compared to regular road segments, traffic flow characteristics at weaving sections are more complicated [4]. For example, in the weaving process, merging and diverging vehicles should enter their target lane by changing lanes in a limited distance at the weaving section without the aid of a traffic control device [5].

Due to urban land constraints, an increasing number of interchanges have been built in China’s metropolitan areas, which greatly reduced travel time and improved traffic capacity. However, as an important part of interchanges, weaving sections have become inclined to be involved in more traffic accidents, such as rear-end and side wipe [6], which have brought about severe casualties and significant economic losses. To reduce traffic conflicts at weaving sections and prevent them from becoming accident-prone locations, why and how traffic crashes happen at weaving sections should be addressed.

Through wide-ranging literature resources, there are several kinds of research streams concentrating on traffic accidents at weaving sections: (i) risk factor analysis of traffic accidents [7,8], (ii) traffic accident distribution characteristics [9,10], (iii) traffic accident prediction [11,12], (iv) traffic accident hazard point identification [13,14], (v) traffic safety assessment [15,16] and (vi) traffic accident prevention [17,18]. However, they failed to compare the effects of various risk factors on different types of traffic accidents at the weaving section and the correlation between traffic accident types and the accident location at weaving sections received little attention. Comprehensive risk analysis has not been completely investigated.

To fill this gap, we will comprehensively compare four types of traffic accidents between associated factors and find the contribution of different risk factors to the four different types of traffic accidents at the weaving section. We divided the weaving section into five zones, which would be helpful in finding hazardous segments for four types of traffic accidents at the weaving section. Traffic accidents can be affected by numerous potential factors, such as drivers’ behavior, weather condition, time, vehicle speed and so on. These factors include continuous and classified variables, which make correlation analysis between risk factors and traffic accidents as a Multivariate Regression (MR) problem with multiple categorical variables. In addition, multinomial logistic regression is a statistical modeling technique on the premise that the probability for a dependent variable is related to a series of potential predictor variables [19], which is widely used to identify factors and predict the likelihood of an outcome [20]. For better understanding of the impact of risk factors on different types of traffic accidents, this research extends the approach from two aspects, namely, (i) identifying six categories of risk factors such as drivers’ attributes, weather conditions, traffic characteristics, driving behavior, vehicle types, and temporal-spatial distribution, (ii) applying the multinomial logistic regression approach to compare four types of traffic accidents between associated factors.

This research makes the following contributions. Firstly, we precisely and comprehensively identify risk factors affecting traffic accidents at the weaving sections. Secondly, we propose a framework to establish the correlation between risk factors and traffic accidents, and compare the impacts of various risk factors on different types of traffic accidents.

The remainder of this paper is organized as follows. Section 2 reviews risk factors of traffic accidents and the application of multinomial logistic regression. Section 3 chooses an observed weaving section, defines five analyzing zones, describes data collection, identifies six categories of risk factors and proposes the multinomial logistic regression as a method used in this research. Section 4 presents the calculation results obtained from the multinomial logistic regression approach. Section 5 analyzes and discusses the results, and the conclusions are drawn in Section 6.

## 2. Literature Review

### 2.1. Risk Factor Analysis of Traffic Accident at Weaving Sections

Because of their serious consequences, traffic safety issues have gained considerable attention from researchers [21,22,23]. As for traffic accidents at the weaving section, important risk factors have been identified. For example, Cirillo analyzed the correlation between accident rates and configuration of weaving section based on the accident data collected in 1961, and the results showed that increasing the length of weaving areas, acceleration lanes, and deceleration lanes can reduce accident rates [24]. Fazio et al. carried out a safety analysis on Interstate 294 in the Chicago metropolitan area, which indicated that lane changing conflicts and following conflicts have a great effect on crash rates at weaving sections [25]. Pulugurtha and Bhatt analyzed the influence of traffic characteristics on crashes using the data of 581 crashes at 25 weaving sections in the Las Vegas metropolitan in 2000. The findings show that crash rates were low with weaving volume less than 15,000 vehicles per day, but high with weaving volume more than 50,000 vehicles per day, and increasing entry volume was the main factor causing the rise of improper lane change [1]. Penmetsa and Pulugurtha found that road features and drivers’ gender were the most two significant factors affecting crash injury severity and the crash frequency at the weaving section using the crash data from 2011 to 2013 obtained from the Highway Safety Information System (HSIS) for the state of North Carolina [26]. Liu et al. studied the safety impacts of lane arrangements at the three types of weaving sections using generalized linear models [27]. Besides the above factors, weather conditions [28,29], vehicle types [30,31] and driving behavior [32,33,34,35] are also considered to be associated with traffic crash risks. In addition, some other researchers studied traffic crash prediction by establishing models to estimate crash likelihood at the weaving section. For instance, Wang et al. utilized a multilevel Bayesian logistic regression model to study crash likelihood using real-time crash data such as crash, geometric, and weather data at the weaving section, which indicated that the distance at which weaving turbulence no longer has impact had the highest risk of the crash [36]. To predict the frequency of accident occurrence, Caliendo et al. employed Poisson distribution and negative multinomial regression models to establish relationships between traffic crashes and traffic flow, geometric infrastructure characteristics and environmental factors [37]. Kiattikomol et al. used a negative binomial regression modeling approach to develop separate models to predict numbers of crashes for different levels of crash severity for interchange segments and non-interchange segments respectively, based on the data obtained from North Carolina, USA [38].

From the previous studies, it is not difficult to conclude that the number of traffic accidents at weaving sections and their injury severity can be affected by different risk factors. However, different groups of risk factors causing traffic crashes were analyzed independently, and there were relatively few studies concentrating on the identification of potential factors associated with classified traffic accidents. Moreover, the correlation between traffic accident types and the accident location at weaving sections was rarely discussed. In view of this, it is necessary to establish a comprehensive risk factors system and the correlation between different types of traffic accidents and their locations at the weaving section should be analyzed.

### 2.2. Application of Multinomial Logistic Regression

MR techniques refer to statistical methods that establish linear or nonlinear quantitative correlations among variables, which are used to discuss the dependence between an outcome and affecting factors [39,40]. An increasing number of studies have focused on MR techniques, such as nonlinear regression [41], linear regression [42], stepwise regression [43], ridge regression [44], lasso regression [45], logistic regression [46] and so on. These methods can effectively solve MR problems from different perspectives. As one of the most applicable logistic regression techniques, multinomial logistic regression can handle the case where the outcome variable is nominal with more than two levels. It is usually utilized to solve MR problems in various aspects, including medicine, economics, engineering and sociology, etc. For example, Kurt et al. studied risk factors affecting coronary artery disease using multinomial logistic regression, which indicated that obesity, smoking status and age were the three most important factors causing coronary artery disease [47]. Lu et al. employed a multinomial logistic model depending upon the estimation of cumulative probability to identify the factors leading to the severity of traffic accidents at Shanghai river-crossing tunnel, and the regression results showed that speed limit and driver’s gender were the two most important factors [48]. To predict company failures, Jabeur applied multinomial logistic regression to find the relationship between bankruptcy and 33 factors for two samples of healthy and failing companies [49].

Despite the wide range of applications of multinomial logistic regression, it is rare in literature to analyze risk factors affecting different types of traffic accidents at the weaving section. This research chooses multinomial logistic regression mainly considering two following advantages: (i) it does not need any assumption about the distribution of variables [50] and (ii) it is suitable for both continuous and categorical variables [51]. Furthermore, because risk factors (independent variables) were classified into six categories and traffic accident types (dependent variables) were divided into four groups (rear-end, side wipe, collision with fixtures and rollover), multinomial logistic regression is a suitable method to solve the regression problem with multiple risk factors and multiple traffic accident types in this research.

## 3. Materials and Methods

### 3.1. Observation of a Weaving Section

In this research, we make a field observation of a weaving section in Xi’an, a city of western China. The observed weaving section is on the 3rd Ring Road between Tian Wang Interchange and Ba Qiao Interchange. The weaving section is 624 m long, which has a three lane mainline with an auxiliary lane and one lane on/off ramps at both ends. Each lane is 3.75 m wide. It is a typical Type A weaving section. Location and configuration of the observed weaving section are illustrated in Figure 2.

### 3.2. Zone Definition

Zone definition of a weaving section is necessary to identify the location of every single accident and obtain the spatial distribution regularities of accidents. Since the concentration of lane changing activity at the weaving section was in the first 200 m, Al-Jameel divided the 200 m segment of the whole M60-J2 weaving section throughout the Greater Manchester area in the UK into 4 equal zones [52]. It is observed that up to 90% of lane changing activities took place in the first 520 m of the observed weaving section. Using Al-Jameel’s definition, we divided the weaving section into five zones including four equal zones and an extra shorter zone in this research, plotted in Figure 3.

Zone 1 (Z1) is the first 130 m from entry point;

Zone 2 (Z2) is from 130 m to 260 m;

Zone 3 (Z3) is from 260 m to 390 m;

Zone 4 (Z4) is from 390 m to 520 m;

Zone 5 (Z5) is the remainder of the weaving section.

### 3.3. Data Collection

Accident data and traffic data are both needed to identify risk factors of traffic accidents. Police accident records are usually adopted as reliable and important sources of traffic accident data [31]. In this research, traffic accident data was extracted from the Traffic Accident Database managed by Xi’an Public Security Bureau. We obtained 768 accident samples of the observed weaving section from 2016 to 2018. Each accident sample includes traffic accident type, drivers’ personal information, vehicle types, accident location features, weather conditions and the time of accidents. The samples included four types of traffic accidents: rear-end, side wipe, collision with fixtures and rollover.

However, police accident records do not contain traffic characteristics when the accident happens. Hence, we employed video surveillance and tachometers to capture traffic data, including traffic volume, vehicle speed, traffic density and weaving ratio. Real-time traffic data was recorded 24 h every day during the three years and was transferred back to the laboratory.

Traffic volume and vehicle speed can be collected directly, and traffic density was calculated using Equation (1).
(1)$$K=\frac{N}{L}$$
where *K* is traffic density (vehicle/km); *N* is the number of vehicles (vehicle); *L* is the length of the lane (km).

Weaving ratio is the percentage of the weaving vehicles out of the total number of the inflow vehicles to the section, which can be calculated using Equation (2).
(2)$$V_{R}=\frac{Q_{W1}{+Q}_{W2}}{Q}$$
where *V_R_* is weaving ratio (%); *Q_W_*_1_ is ramp-to-mainline traffic volume (vehicle); *Q_W_*_2_ is mainline-to-ramp traffic volume (vehicle); *Q* is the total traffic volume in the weaving section (vehicle).

### 3.4. Risk Factors

Appropriate identification of risk factors affecting traffic accidents is necessary. From the existing research, driver attributes [53], weather conditions [54], traffic characteristics [55], driving behavior [56], temporal-spatial distribution [57] and vehicle types [58] can all influence the possibility of a traffic accident. We established the risk factor system based on the literature plus two newly added risk factors, i.e., weaving ratio and accident location. The risk factor system consists of six categories including 16 risk factors as follows.

#### 3.4.1. Drivers’ Attributes

Drivers’ age and gender are always considered as important factors in the research of traffic accidents. According to different driving behaviors and skills, drivers’ age is classified into four groups: ≤25, 26–44, 45–64 and ≥65. Additionally, male and female drivers also have a different driving preference.

#### 3.4.2. Weather Conditions

Weather condition is an external contributor to traffic accidents, especially some extreme climatic conditions. From the data collection, eight different weather conditions were observed, i.e., sunny, cloudy, rainy, snowy, foggy, windy, dusty, hail [54].

#### 3.4.3. Traffic Characteristics

We used two main traffic flow parameters, i.e., traffic density and weaving ration to represent traffic characteristics in the weaving section. Traffic density is grouped into four categories: ≤10 vehicle/100 m, 11–20 vehicle/100 m, 21–30 vehicle/100 m and ≥31 vehicle/100 m. Weaving ration is also divided into four groups: ≤10%, 11–25%, 26–40% and ≥41%.

#### 3.4.4. Driving Behavior

Speed is an important risk causing traffic accidents, which is classified as ≤50 km/h, 51–80 km/h, 81–100 km/h and ≥101 km/h according to their different possibilities of causing an accident. At a weaving section, lane change is a common driving behavior in ramp-to-mainline and mainline-to-ramp traffic flows, which easily leads to side wipe traffic accidents.

#### 3.4.5. Vehicle Types

From the 768 accident samples of the observed weaving section, five types of vehicles were involved in traffic accidents, i.e., private car, minibus, bus, taxi and truck.

#### 3.4.6. Temporal-Spatial Distribution

Seasons are grouped into four periods: spring, summer, autumn and winter. Time is divided into five categories: 00:00–06:59 a.m., 07:00–08:59 a.m., 09:00 a.m.–16:59 p.m., 17:00–19:59 p.m. and 20:00–23:59 p.m. 07:00–08:59 a.m. is the morning rush hour, and 17:00–19:59 p.m. is the evening rush hour. Weekends and weekdays are also classified. According to the zone definition of the weaving section, accident location is divided into five zones: Zone 1, Zone 2, Zone 3, Zone 4, and Zone 5.

### 3.5. Methods

Considering the advantages presented in Section 2.2, the multinomial logistic regression is employed in this research, which is formulated in detail as follows.

It denotes that *X_i_* is the variable of traffic accident affecting factor *i*, *i* = 1, 2, …, *m*, *P_j_* is the probability of traffic accident type *j*, *j* = 0, 1, ⋯, *n*−1. j = 0 is used as the referent type of traffic accident. The regression relationship between *X_i_* and *P_j_* is formulated as.
(3)$$P_{j}=\frac{{\exp(\alpha }_{j}+\sum_{i=1}^{m}\beta_{ij}\cdot X_{i})}{{1+\exp(\alpha }_{j}+\sum_{i=1}^{m}\beta_{ij}\cdot X_{i})},j=1,2,\cdots ,n$$
where *m* is the number of risk factors; *n* is the number of traffic accident types. *P_j_* must be constrained as:(4)$$\overset{n-1}{\underset{j=1}{\sum }}P_{j}=1$$

Denote $Y_{j}=\alpha_{j}+\beta_{1j}\cdot X_{i}+\beta_{2j}\cdot X_{i}+\cdots +\beta_{mj}\cdot X_{i}=\alpha_{j}+\sum_{i=1}^{m}\beta_{ij}\cdot X_{i}$.

Where $Y_{j}$ is the total discrimination value which reflects the quantitative characteristics of *i*th traffic accident affecting factor; $\beta_{ij}$ is the coefficient which reflects the degree of relevant independent variables *X_i_*; $\alpha_{j}$ is a constant.

Then, Equation (3) can be rewritten as:(5)$$P_{j}=\frac{\exp\left(Y_{j}\right)}{1+\exp\left(Y_{j}\right)}$$

From Equation (5), $\exp\left(Y_{j}\right)$ can be obtained as:(6)$$\exp\left(Y_{j}\right)=\;\frac{P_{j}}{1-P_{j}}$$

The natural logarithm is taken on Equation (6), and $Y_{j}$ can be formulated as:(7)$$Y_{j}=\ln\frac{P_{j}}{1-P_{j}}{\;=\;\alpha }_{j}+\sum_{i=1}^{m}\beta_{ij}\cdot X_{i}$$

We use odd ration (OR) to estimate the effects of different traffic accident affecting factors on the possibility of traffic accident types. OR can be computed by Equation (8).
(8)$${OR}_{ij}{\;=\;\mathrm{EXP}(\beta }_{ij})$$
where the odd ration of *i*-th traffic accident affecting factor to traffic accident type *j*, which indicates the relative amount by which the odds of traffic accidents increases (OR > 1) or decreases (OR < 1) when the value of affecting factors increase per unit.

For the above multinomial logistic regression model, the significance of variables should be assessed. We employed the Wald test to achieve the significance testing, which is defined as *β*/SE (standard error).

## 4. Results

The multinomial logistic regression model was applied using the collected data described in Appendix A, Table A1, which shows the number of four types of traffic accidents associated with every single factor. Table 1 displays the multinomial logistic regression results, which indicates that the affecting factors have different effects on the four traffic accident types. The detailed description of results is as follows.

### 4.1. Rear-End

Male drivers are 1.821 times more likely to involve in rear-end than female drivers. Compared to young drivers (age ≤25), drivers of age 26–44 and ≥65 have a higher risk of rear-end, but the age group of 45–64 has the lowest risk. Snow and fog are the two most significant weather conditions, which are more inclined to cause rear-end than other weather conditions. The higher the traffic density is, the higher the risk it will have. It seems that weaving ratio does not have an obvious impact on rear-end, because the ORs show little variation with the change of weaving ratio. Traffic density of ≥31 vehicle/100 m is associated with the highest possibility of rear-end (OR = 6.321). In addition, lane change behavior is 2.143 times more probably to bring about rear-end than no lane change behavior. Private cars, taxis and trucks are more inclined to be involved in rear-end, especially trucks. Compared to other seasons, more rear-end traffic accidents occur in winter. As for time, morning rush hour (07:00–08:59) exhibits the highest risk for rear-end, while the time period 00:00–06:59 has the lowest risk. Rear-end is less likely to occur during weekends. It is found that Zone 4 and Zone 5 correlated with a higher chance of rear-end at the weaving section.

### 4.2. Side Wipe

Male drivers are more inclined to cause a side wipe than female drivers. Compared to young drivers (age ≤25), drivers of age ≥65 are 1.427 times more likely to be involved in a side wipe, but the other two age groups have lower risks. Rain is found to be the most likely condition to lead to a side wipe compared to the other weather conditions. There is a positive correlation between traffic density and side wipe risk. The higher the weaving ratio is, the higher possibility the side wipe will have. Weaving ration of ≥41% is the most likely to bring about a side wipe (OR = 7.512). Furthermore, compared to low speed, higher speed has higher chances of a side wipe, but middle speed 51–80 km/h has the highest risk. Lane change is also an important factor affecting the side wipe. Private cars and taxis are more associated with side wipes, while trucks are less likely to be involved in the side wipe. Side wipe is more likely to happen in summer and in the evening rush hour (17:00–19:59), but less likely at weekends. There is a higher risk of side wipe in Zone 2 and Zone 3 at the weaving section.

### 4.3. Collision with Fixtures

Interestingly, unlike the other traffic accident types, female drivers have a higher possibility of collision with fixtures than male drivers. Drivers of age 26–44 and ≥65 have a higher risk of collision with fixtures, but drivers of age 45–64 have a lower risk compared to young drivers (age ≤25). Snowy weather has the highest possibility to bring about the collision with fixtures (OR = 6.137). Higher traffic density will reduce the probability of collision with fixtures. Risk of collision with fixtures becomes higher as the weaving ratio increases, but the ORs show little variation. It is obtained that when speed is more than 101 km/h, vehicles have the highest risk of being associated with the collision with fixtures compared to other vehicle speed categories. Lane change behavior is also an important factor causing the collision with fixtures. Private cars are more inclined to be involved in a collision with fixtures. A collision with fixtures is more likely to occur in winter and in the time period 09:00–16:59, but less likely at weekends. Zone 2 and Zone 5 of the weaving section are associated with a higher chance of collision with fixtures than the other three zones.

### 4.4. Rollover

Male drivers are 2.013 times more likely to be involved in the rollover. Drivers of age 26–44 are more inclined to cause a rollover, but drivers of age 45–64 and ≥65 have lower risks compared to young drivers (age ≤25). Rainy weather has the greatest impact on rollover (OR = 4.871). Higher traffic density shows a lower probability of rollover. Middle weaving ratio (26–40%) has a higher risk of rollover, while lower weaving ratio (11–25%) and higher weaving ratio (≥41%) are correlated with lower risk. Risk of rollovers increases with vehicle speed. Lane change indicates a much higher risk of rollover than no lane change. Private cars are more inclined to be involved in the rollover, but trucks present a lower risk of rollover. It is found that rollover is more likely to occur in summer, especially in winter. The time period of 20:00–23:59 has the highest risk of rollover, while the time period of 17:00–19:59 shows the lowest risk. In addition, rollover is more likely to happen during weekdays than weekends. There is a higher risk of rollover in Zone 2 and Zone 5 at the weaving section.

## 5. Discussion

### 5.1. Drivers’ Attributes

Drivers’ gender is found to be an important factor affecting traffic accidents at the weaving section [59]. From the statistical analysis, we know that male drivers have a higher possibility of involvement in rear-end, side wipe and rollover than female drivers, but for collision with fixtures, female drivers have higher risks than male drivers.

According to the results, drivers aged 26–44 and ≥65 are more inclined to be involved in rear-end, side wipe and rollover at the weaving section. This is probably because drivers aged 26–44 usually have bad driving behaviors such as speeding, using cell phones while driving, frequent lane change, etc. and drivers aged ≥65 have a decline of driving skills.

Regarding this concern, the criteria for driving license issuance for older drivers requires further analysis and stricter measures should be adopted to prevent drink driving, distracted driving, road rage, etc.

### 5.2. Weather Conditions

Bad weather conditions are always identified as a high risk, which may lead to a severe traffic accident injury. Compared to sunny days, other weather conditions have higher risks to a different extent at the weaving section. Snow has the highest possibility of rear-end and collision with fixtures, which is probably due to the slippery pavement. While rain has the highest risk of side wipe and rollover, because rain has a great impact on drivers’ vision and sight.

As a response, drivers are advised to drive less and not to drive faster than the speed limit during bad weather conditions. Additionally, drivers should check their tires regularly to ensure the grip of the tires.

### 5.3. Traffic Characteristics

According to the results, higher traffic density is more associated with rear-end and side wipe, because higher traffic density means a shorter average space headway and more congested traffic flow. There is an inverse relationship between traffic density and the risk of collision with fixtures, as well as rollover. Weaving ratio is an important parameter to describe the characteristics of traffic flow at the weaving section, which has a great risk of side wipe and rollover, but has little effect on rear-end and collision with fixtures.

Accordingly, it is advisable for drivers to keep a safe distance from vehicles in front of them, and prepare for a lane change ahead of time when they enter the ramp from mainline and vice versa. The aggressive competition for a lane-changing opportunity is forbidden at the weaving section.

### 5.4. Driving Behavior

Many existing research results show that speeding and frequent lane change are the two most common dangerous driving behaviors, which cause most of the traffic accidents. From the results, it is revealed that higher vehicle speed has a higher chance of causing rear-end, collision with fixtures and rollover, but has a lower risk of side wipe. Lane change has a great effect on side wipe and rollover at the weaving section. Hence, it is necessary to promote mandated speed limits and no overtaking at the weaving section and to build more than one auxiliary lane for the weaving section if possible.

### 5.5. Vehicle Types

Private cars account for 87% of the total traffic volume at the observed weaving section, which has the highest possibility of involvement in traffic accidents, i.e., they are 4.717 times more likely to be involved in rear-end, 5.435 times in side wipe, 3.612 times in collision with fixtures and 5.498 times in rollover than non-private cars. Because trucks always have a longer braking distance, they are more associated with rear-end compared to the other three types of traffic accidents at the weaving section. Since taxis usually have a high speed and frequent lane change, they are also often at risk of traffic accidents.

In view of this, traffic regulations could be considered to separate passenger cars and trucks. In addition, some heavy trucks may only be allowed to drive along specialized lanes or during a certain times at the weaving section.

### 5.6. Temporal-Spatial Distribution

Rear-end, collision with fixtures and rollover are more inclined to happen in winter, because bad weather conditions such as rain, snow, fog and hail often occur in winter, which are risks for traffic accidents, while side wipe is more likely to happen in summer. Because traffic density is usually high in rush hours, morning rush hour often has a higher risk of rear-end, but it is found that evening rush hour has a higher risk of side wipe. Collision with fixtures and rollover occur in the period 20:00–23:59 most often, which is probably due to drowsy driving. All four types of traffic accidents are much more likely to happen during weekdays than weekends. Traffic accidents have significant spatial distribution characteristics at the weaving section in this research, which is rarely studied in the previous literature. It is found that rear-end is more likely to occur in Zone 4 and Zone 5, side wipe has a higher possibility of happening in Zone 2 and Zone 3, collision with fixtures and rollover are more inclined to happen in Zone 2 and Zone 5.

As a result, traffic demand management policies such as flexible work time should be designed to shift traffic from peak periods. Extension of the weaving length, improvement of geometric conditions, avoiding horizontal curve or vertical curve, etc. are also should be considered to enhance the safety level of the weaving section.

## 6. Conclusions

To solve the problem of land shortage, lots of interchanges have been built in China’s metropolitan areas, which increased transport mobility greatly. As an important configuration of interchanges, weaving section is likely to be involved in more traffic accidents, which may bring about severe casualties and significant economic losses. Analysis of risk factors has become necessary to prevent various types of traffic accidents at weaving sections. In view of this, this research established a risk factor identification and analysis framework of traffic accidents at weaving sections using multinomial logistic regression. Correlation between six categories of 16 risk factors (drivers’ attributes, weather conditions, traffic characteristics, driving behavior, vehicle types and temporal-spatial distribution) and four types of traffic accidents (rear-end, side wipe, collision with fixtures and rollover) was identified based on 768 accident samples of an observed weaving section in Xi’an, China from 2016 to 2018. Different significance of 16 risk factors in the four types of traffic accidents was compared. The main results are concluded as follows:(1)Traffic accidents at the weaving section are mainly affected by factors such as drivers’ gender and age, weather condition, traffic density, weaving ratio, vehicle speed, lane change behavior, private cars, season, time period, day of week and accident location, but these factors have different effects on the four traffic accident types.(2)Traffic density of ≥31 vehicle/100 m has the highest risk of causing rear-end, weaving ration of ≥41% has the highest possibility of bringing about a side wipe, collision with fixtures is the most likely to happen in snowy weather, and rollover is the most likely to occur in rainy weather.

Due to their serious results, traffic accidents attract tremendous attention from researchers. However, few studies have yet analyzed and compared the different significance of risk factors in different types of traffic accidents at the weaving section. This research intends to provide a reference to improve the safety of weaving sections in China. However, there are two limitations in this research. Firstly, this research focuses on the traffic accidents at the Type A weaving section, but does not consider other types of weaving sections (Types B and C), which have different configuration characteristics. Secondly, the injury severity of traffic accidents was not involved in this research. These limitations will be taken into account in the following studies.

## Figures and Tables

**Figure 1 ijerph-16-01542-f001:**
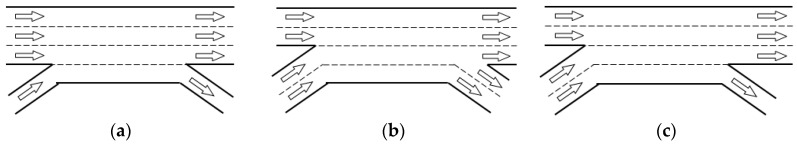
Three types of weaving sections. (**a**) Type A weaving section; (**b**) Type B weaving section; (**c**) Type C weaving section.

**Figure 2 ijerph-16-01542-f002:**
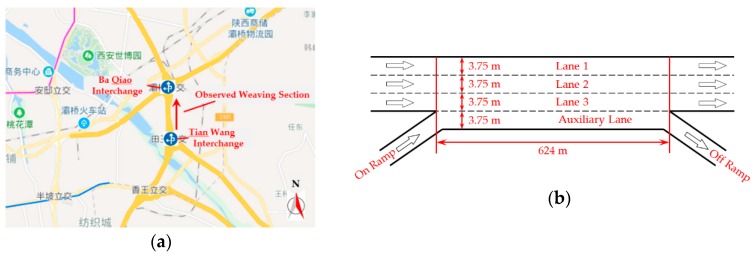
The observed weaving section. (**a**) Location of the observed weaving section; (**b**) Configuration of the observed weaving section.

**Figure 3 ijerph-16-01542-f003:**
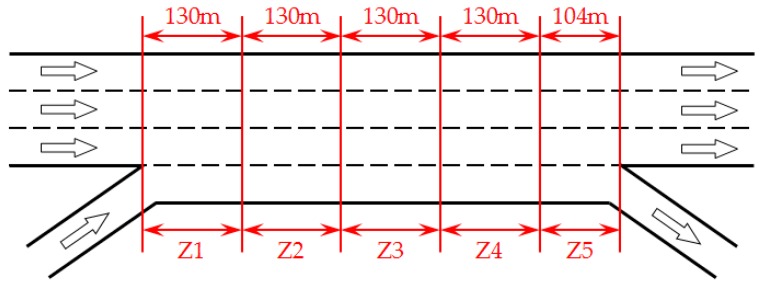
Division of the observed weaving section into five zones.

**Table 1 ijerph-16-01542-t001:** Multinomial logistic regression results.

Factors		Rear-End			Side Wipe			Collision with Fixtures			Rollover		
		*β*	Wald	OR	*β*	Wald	OR	*β*	Wald	OR	*β*	Wald	OR
Gender (Reference: Female)	Male	−6.229	4.124	1.821	−8.036	1.543	1.732	−6.861	2.653	0.811	−7.231	2.473	2.013
Age (Reference: ≤25)	26–44	−9.137	11.763	1.123	−10.51	2.854	1.265	−9.618	6.685	1.521	−9.899	6.065	2.223
	45–64	−4.215	6.432	0.743	−4.814	0.894	0.548	−4.425	3.275	0.682	−4.547	2.891	0.873
	≥65	−9.879	5.447	1.113	−11.43	5.834	1.427	−10.42	5.668	1.652	−10.74	5.695	0.728
Weather Conditions (Reference: Sunny)	Cloudy	−13.98	16.543	1.004	−15.26	5.983	1.112	−14.42	10.52	1.071	−14.69	9.789	1.239
	Rainy	13.74	8.762	2.687	11.48	6.936	3.431	12.95	7.721	1.652	12.49	7.594	4.871
	Snowy	−17.13	2.871	5.432	−22.05	4.991	2.721	−18.85	4.079	6.137	−19.86	4.227	4.652
	Foggy	−9.725	1.762	4.247	−13.53	3.746	2.689	−11.06	2.893	2.148	−11.84	3.031	2.651
	Windy	−13.25	1.983	2.324	−14.85	9.431	1.436	−13.81	6.228	1.872	−14.14	6.747	1.625
	Dusty	−10.13	4.672	1.872	−11.92	5.983	1.673	−10.76	5.419	1.562	−11.12	5.513	1.238
	Hail	−11.01	6.432	1.004	−12.27	6.783	1.121	−11.45	6.632	1.105	−11.71	6.656	1.217
Traffic Density (Reference: ≤10 vehicle/100m)	11–20 vehicle/100 m	10.75	11.325	2.315	8.461	4.093	2.422	9.951	7.203	0.621	9.482	6.699	1.012
	21–30 vehicle/100 m	−6.145	4.761	4.332	−9.84	1.119	3.621	−7.438	2.685	0.341	−8.194	2.432	0.535
	≥31 vehicle/100 m	−6.533	2.431	6.321	−13.73	1.329	7.123	−9.053	1.803	0.214	−10.53	1.726	0.322
Weaving Ratio (Reference: ≤10%)	11–25%	−5.636	2.984	1.013	−7.358	3.095	1.654	−6.239	3.047	1.012	−6.591	3.055	0.873
	26–40%	3.758	4.872	1.103	0.26	0.678	3.543	2.534	2.481	1.033	1.818	2.189	2.451
	≥41%	9.72	2.743	1.024	2.325	7.774	7.512	7.132	5.611	1.154	5.62	5.961	0.231
Vehicle Speed (Reference: ≤50 km/h)	51–80 km/h	−7.986	1.834	1.533	−9.394	4.875	2.312	−8.479	3.567	1.276	−8.767	3.779	1.467
	81–100 km/h	−7.08	1.336	3.121	−8.808	1.076	1.643	−7.685	1.188	3.567	−8.038	1.175	3.543
	≥101 km/h	−15.87	2.991	5.643	−18.07	4.095	1.012	−16.64	3.626	6.543	−17.09	3.697	7.113
Lane Change (Reference: No Lane Change)	Changing lanes	−15.08	5.982	2.143	−24.39	3.775	7.124	−18.34	4.724	6.641	−20.24	4.571	7.214
Private Car (Reference: No private car involved)	Private car involved	−16.26	11.437	4.717	−17.89	9.453	5.435	−16.84	10.31	3.612	−17.17	10.17	5.498
Minibus (Reference: No minibus car involved)	Minibus car involved	−4.989	0.763	0.887	−6.961	1.734	1.912	−5.679	1.316	1.654	−6.082	1.384	1.784
Bus (Reference: No bus involved)	Bus involved	2.876	1.218	0.723	1.7	0.673	1.211	2.464	0.907	1.109	2.224	0.869	1.045
Taxi (Reference: No taxi involved)	Taxi involved	3.214	0.746	1.234	2.139	0.943	1.114	2.838	0.858	1.092	2.618	0.872	1.218
Truck (Reference: No truck involved)	Truck involved	−8.944	2.198	4.019	−11.17	4.874	0.617	−9.722	3.723	2.215	−10.18	3.909	0.485
Seasons (Reference: Spring)	Summer	−6.236	1.043	1.012	−7.346	4.098	2.035	−6.625	2.784	1.108	−6.852	2.997	1.715
	Autumn	−6.345	4.15	1.123	−7.608	3.004	1.187	−6.787	3.497	1.165	−7.045	3.417	0.898
	Winter	−8.667	6.437	2.102	−10.82	2.945	1.051	−9.421	4.447	1.986	−9.862	4.204	2.016
Time (Reference: 00:00-06:59)	07:00–08:59	−11.16	4.983	2.451	−14.06	5.843	1.762	−12.18	5.473	1.263	−12.77	5.533	0.832
	09:00–16:59	−10.37	2.336	1.873	−12.43	6.657	1.932	−11.09	4.799	1.784	−11.51	5.104	1.073
	17:00–19:59	5.613	4.776	1.763	3.737	1.002	1.943	4.956	2.625	1.672	4.573	2.362	0.733
	20:00–23:59	6.554	8.984	1.032	5.457	1.431	1.176	6.17	4.679	2.154	5.946	4.153	2.149
Day of Week (Reference: Weekends)	Weekdays	10.64	2.119	4.732	6.258	2.843	4.512	9.108	2.532	3.872	8.211	2.582	4.034
Accident Location (Reference: Zone 1)	Zone 2	−17.82	1.054	1.121	−26.46	1.564	8.435	−20.84	1.345	4.02	−22.61	1.382	3.831
	Zone 3	−9.143	2.541	1.638	−13.27	4.438	4.014	−10.59	3.622	1.836	−11.43	3.754	0.843
	Zone 4	−7.257	1.774	4.325	−8.801	4.663	1.457	−7.797	3.421	0.667	−8.113	3.622	1.103
	Zone 5	−8.642	1.054	3.212	−11.17	3.317	2.426	−9.528	2.344	0.784	−10.05	2.501	4.037

Note: Significant at 5% level.

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
