# Peer review of "Risk Factors Affecting Traffic Accidents at Urban Weaving Sections: Evidence from China"

_ijerph, 2019, doi:10.3390/ijerph16091542_

Round 1

Reviewer 1 Report

The article deals with an important issue from the point of view of road safety and studies the case related to the weaving sections, some of the most dangerous segments on multi-lane roads.

As recommendations for improvement, I suggest the following points:

- describe types A, B and C of weaving sections.

- incorporate proposals for public policies or specific action in the sections studied, based on the results obtained. How can road safety be improved in this case study and in general in the light of the results?

- increase the conclusions with a summary of the results obtained from the discussions, not just the contextualization, methodology used and limitations and future work.

With these minor changes, I recommend the publication of this manuscript in International Journal of Environmental Research and Public Health.

Author Response

Point 1: The article deals with an important issue from the point of view of road safety and studies the case related to the weaving sections, some of the most dangerous segments on multi-lane roads. As recommendations for improvement, I suggest the following points.

Response 1: The authors greatly appreciate the reviewer’s encouragement and suggestions. This memo documents our responses to all review comments. The appropriate changes have been made to the manuscript.

Point 2: Describe types A, B and C of weaving sections.

Response 2: The authors have provided descriptions of the three types of weaving sections. Please refer to Lines 36-42. In addition, a figure was given to indicate the characteristics of the three types of weaving sections. Please refer to Figure 1.

Point 3: Incorporate proposals for public policies or specific action in the sections studied, based on the results obtained. How can road safety be improved in this case study and in general in the light of the results?

Response 3: The authors have provided some policy implications according to every point of discussions. Please refer to Lines 335-337, Lines 344-346, Lines 354-356, Lines 362-364, Lines 373-375, Lines 388-391.

Point 4: Increase the conclusions with a summary of the results obtained from the discussions, not just the contextualization, methodology used and limitations and future work.

Response 4: The authors have added a summary of the research results into Conclusions. Please refer to Lines 405-412.

Point 5: With these minor changes, I recommend the publication of this manuscript in International Journal of Environmental Research and Public Health.

Response 5: The authors are thankful to the reviewer’s great job again.

Reviewer 2 Report

The paper entitled "Risk Factors Affecting Traffic Accidents at Urban Weaving Section: Evidence from China" provides a well-referenced and structured content that contributes to analysing accidents at weaving sections. The nearly 60 references significantly support the understanding. Nevertheless, the authors should exclude mentioning the used software, SPSS, which entirely does not connect to the previous content. It is important to highlight the approach as software-independentent and reproducible. Instead of the formulations in lines 250 to 254, there should be a proper overview on subsections 4.1 to 4.4.

Author Response

Point 1: The paper entitled "Risk Factors Affecting Traffic Accidents at Urban Weaving Section: Evidence from China" provides a well-referenced and structured content that contributes to analysing accidents at weaving sections. The nearly 60 references significantly support the understanding.

Response 1: The authors greatly appreciate the reviewer’s encouragement and suggestions. The authors have made an appropriate revision to the manuscript.

Point 2: The authors should exclude mentioning the used software, SPSS, which entirely does not connect to the previous content. It is important to highlight the approach as software-independentent and reproducible.

Response 2: The authors have removed the content related to the software SPSS. Please refer to Lines 267-268. Meanwhile, the authors gave more details to the approach.Please refer to Lines 255-259.

Point 3: Instead of the formulations in lines 250 to 254, there should be a proper overview on subsections 4.1 to 4.4.

Response 3: The authors have revised the section 4 and provided a proper overview. Please refer to Section 4 (Lines 267-326).
